# The Effects of Simulation Problem-Based Learning on the Empathy, Attitudes toward Caring for the Elderly, and Team Efficacy of Undergraduate Health Profession Students

**DOI:** 10.3390/ijerph18189658

**Published:** 2021-09-14

**Authors:** Hae-Kyoung Son

**Affiliations:** College of Nursing, Eulji University, Seongnam 13135, Korea; sonhk@eulji.ac.kr; Tel.: +82-31-740-7157; Fax: +82-31-740-7359

**Keywords:** nursing, dental hygiene, simulation problem-based learning, elderly care, empathy, team efficacy

## Abstract

Undergraduate students studying health professions receive a uniprofessional education in an isolated educational environment within the university curriculum, and they have limited opportunities to experience collaborative learning through interactions with other professions. This study adopted a one-group, pretest–posttest, quasi-experimental design to investigate the effect of an undergraduate course that applied simulation problem-based learning (S-PBL) on nursing and dental hygiene students’ empathy, attitudes toward caring for the elderly, and team efficacy. The S-PBL was designed based on the ARCS model of motivation proposed by Keller, and the subjects (n = 24) participated in a small group activity of identifying and checking for medical errors that may pose a threat to patients’ safety. The results showed that there was a statistically significant increase in the subjects’ attitudes toward caring for the elderly (t = 3.11, *p* = 0.01) and team efficacy (t = 2.84, *p* = 0.01) after participating in the S-PBL. The teaching method developed by this study aims to counteract the problems of the limited experience available to undergraduate health profession students during clinical practicum in the context of the COVID-19 pandemic and the limitations of interprofessional education, and it has established the groundwork for further exploration of the learning transfer effect of S-PBL.

## 1. Introduction

The discipline of health profession studies underlines the value of interpersonal relationships. A range of competencies are required to build trust and promote interactions in relationships between health care service providers and therapeutic relationships with patients [1,2]. However, undergraduate students of health professions, including nursing and dental hygiene, have received a uniprofessional education in an isolated educational environment within the university curriculum, and they have limited opportunities to experience collaborative learning through interactions with other professions [3,4]. Therefore, preparing and developing an interprofessional education (IPE) curriculum can be challenging to participate in and facilitate curricular activities that involve different types of interactive learning [5,6]. Some of the main challenges to IPE are crowded timetables and simultaneous movement of students and faculties for similar classes among other health professions [6].

Due to the coronavirus disease (COVID-19) in particular, universities have been forced to hold non-face-to-face classes, and medical institutions have either not permitted clinical practice or have operated their training on a restricted basis [7,8]. Ultimately, the difficulties of experiencing direct interactions with various health professions in clinical settings have been compounded [7,8]. In addition, patients’ symptoms have been increasing in variety, complexity, and severity, requiring close collaboration between health professions to derive optimal clinical outcomes [9,10]. In other words, improving teamwork and increasing cooperation in healthcare services is needed [11]. Interprofessional collaboration of health disciplines affords patients an opportunity to receive quality care and successful patient-oriented healthcare service that has been influenced by other health disciplines and expertise [5]. Therefore, there has been mounting attention drawn to the necessity of IPE in various health profession studies, such as nursing and dental hygiene, and the promotion of the learning transfer effect of simulation education [4,12].

Several studies have presented evidence that simulation-based learning is effective for enhancing learners’ knowledge, skills, critical thinking, and satisfaction. As the use of simulation is recommended for patients’ safety, it can be employed as an effective educational method in various health professions, including nursing. In particular, simulation problem-based learning (S-PBL) uses manikins, simulators, and standardized patients in an environment similar to the actual clinical settings of health care services to provide learners with opportunities for repeated learning, immediate feedback, evaluation, and reflection, thus serving as a useful educational strategy for the integration of theoretical and practical learning [13,14]. S-PBL also employs peer role-play, rendering it a suitable methodology for experiencing and training in interpersonal relationships between various health professions and contextual communication. Thus, IPE-based S-PBL is a pedagogical method which provides a simulation-based education based on an interprofessional collaboration of health disciplines.

Under the novel conditions of the COVID-19 pandemic, the development, application, and verification of the effects of S-PBL within the university curriculum (as a teaching method that can supplement the limited experience and limitations of IPE in the clinical practicum of health profession undergraduates) has significance in terms of education, research, and clinical practice. Considering that the pandemic has hindered communication and teamwork between health professions and students in the clinical setting [8], this exploratory study applies IPE-based S-PBL to syllabus development, and it highlights opportunities for communication, collaboration, and interaction between health professions. In this way, it aims to examine the impact of S-PBL application on the empathy, attitudes toward the elderly, and team efficacy of health profession undergraduates, in terms of the learning transfer promotion facilitated by S-PBL. 

Since empathy in the health professions has been reported as a factor that improves clinical outcomes, the application of a variety of educational strategies that can enhance empathy, as well as the verification of their effectiveness, is instrumental [15]. Empathy within the health professions is defined as the multi-dimensional understanding of the emotive, moral, cognitive, and behavioral dimensions of patients’ emotions and perspectives, and the ability to approach the concept from the clinical context is necessary [15]. According to the educational objectives of S-PBL based on IPE, in addition to increasing the learner’s knowledge, the improvement of attitudes toward the elderly, and team efficacy based on communication between health professions, it also has significance in terms of providing basic data for promoting students’ learning transfer. In particular, team-based learning in S-PBL is an education strategy based on a team system in which members share a vision, build efficient communication, interact with each other, and achieve goals [16]. Therefore, S-PBL combined with team-based learning could represent an effective methodology for improving undergraduate health profession students’ positive educational outcomes, including their communication skills, nursing performance confidence, team performance, and team efficacy [17]. This study investigates the effect of IPE-based S-PBL on the empathy, attitudes toward caring for the elderly, and team efficacy of health profession undergraduates, thus presenting an educational foundation from which to provide high-quality services in a complex health care system.

### 1.1. Aims

This study investigates the effect of an S-PBL applied curriculum that focuses on the empathy, attitudes toward caring for the elderly, and team efficacy of undergraduate health profession students. The research specifically focused on the following. First, undergraduates’ existing levels of empathy, attitudes toward caring for the elderly, and team efficacy were assessed. Second, the correlation between empathy, attitudes toward caring for the elderly, and team efficacy among the undergraduates was examined. Third, the effect of S-PBL on empathy, attitudes toward caring for the elderly, and team efficacy of undergraduates was confirmed.

### 1.2. Research Hypotheses

This study proposed the following hypotheses:

**Hypothesis** **1** **(H1).**
*Health profession undergraduates’ empathy will be improved after participating in S-PBL compared with before participation.*


**Hypothesis** **2** **(H2).**
*Health profession undergraduates’ attitudes toward caring for the elderly will be improved after participating in S-PBL compared with before participation.*


**Hypothesis** **3** **(H3).**
*Health profession undergraduates’ team efficacy will be improved after participating in S-PBL compared with before participation.*


## 2. Materials and Methods

### 2.1. Research Design

This study employed a one-group, pretest–posttest, quasi-experimental design. The research design is presented in Figure 1 below.

### 2.2. Subjects and Setting

The study’s subjects were students enrolled in and attending the departments of nursing and dental hygiene at a college specializing in health care in S City (a metropolitan area) in South Korea. The sample included students who wished to voluntarily participate in a major elective course called “Senior Care Convergence Clinical Practicum” and had completed the basic IPE courses “Basic Understanding of Senior Care” and “Basic Senior Care Convergence” as prerequisite subjects before participating in the S-PBL. To calculate the appropriate number of subjects, G*Power Software (G*Power 3.1.7, Heinrich-Heine-University, Düsseldorf, Germany) was used, with a significance level α of 0.08, a power of 0.95, and an effect size of 0.8 [18]. Consequently, the necessary minimum number of the sample was set at 23; considering a dropout rate of 10%, convenience sampling of a total of 27 subjects was conducted. Three subjects dropped out due to personal reasons, such as absence due to illness; therefore, a total of 24 subjects were included in the final analysis.

### 2.3. Outcome Assessment

#### 2.3.1. General Characteristics

The participants’ general characteristics were recorded, including their ages, genders, majors, grades, religions, health status, motivations for major choice, academic achievements, residential type, and satisfaction with their major.

#### 2.3.2. Empathy

The Korean version of Davis’ [19] Interpersonal Reactivity Index IRI (K-IRI), whose reliability and validity were verified by Kang, Kee, Kim, Jeong, Hwang, Song, and Kim [20], was used to assess the participants’ empathy. This tool consists of four subscales: perspective taking (PT) and fantasy (FS) (which comprise cognitive empathy) and empathic concern (EC) and personal distress (PD) (which comprise emotional empathy). The tool includes a total of 28 items (seven in each subscale), and each item is responded to via a 5-point scale ranging from “The statement does not describe me well” (0 points) to “The statement describes me well” (4 points). Negative items were scored in reverse. The higher the total score, the higher the participant’s level of empathy. In terms of reliability, Cronbach’s a has been reported as 0.80 for the overall scale, 0.61 for perspective taking, 0.81 for fantasy, 0.73 for empathic concern, and 0.71 for personal distress [20]. In the present study, Cronbach’s a was 0.83 for the overall scale, 0.65 for perspective, 0.85 for fantasy, 0.82 for empathic concern, and 0.84 for personal distress.

#### 2.3.3. Attitudes Toward Caring for the Elderly

Attitudes toward caring for the elderly refers to the emotions felt about the elderly and the positive or negative responses displayed when caring for them. This was assessed using Sanders, Montgomery, Pittman, and Balkwell’s Aging Semantic Differential Scale [21] and the Attitude Toward the Elderly scale revised by Joung & Hyun [22], and it was based on Maxwell and Sullivan’s Empathy and Attitude Toward the Elderly Questionnaire [23]. This measure uses a 5-point Likert scale for responses with a total of 17 items; the total score ranges from 17 (lowest) to 85 points (highest). A higher total score indicates a more positive attitude toward the elderly. In terms of the reliability of the scale, Cronbach’s a was 0.92 in Joung & Hyun’s research [22] and 0.93 in the present study.

#### 2.3.4. Team Efficacy

Team efficacy was assessed with an instrument developed by Marshall [24] and modified and supplemented by Kwon [25]. This instrument includes a total of eight items, and the response to each item is given using a 5-point Likert scale from 1 point (“strongly disagree”) to 5 points (“strongly agree”). A higher total score indicates higher team efficacy. In terms of the scale’s reliability, Cronbach’s a was 0.97 in Kwon’s research [25] and 0.94 in the present study.

### 2.4. Interventions: S-PBL

S-PBL based on IPE is designed so that the learner’s motivation is induced by interactions between attention, relevance, confidence, and satisfaction, the four factors of the ARCS model of motivation proposed by Keller [26] (Figure 2). Previous studies have reported improvement in learners’ motivation and attitude following the application of the ARCS model in different courses for various age groups, from elementary school to undergraduate students [27]. 

In the S-PBL applied in this study, in the first subcategory of Attention (A), the activities began with introductory icebreakers in a small group. This was followed by a presentation of audio–visual materials such as PowerPoint files (e.g., an upside-down word in a visual) and animations to provide guidance and learning materials, as well as create a story, in order, after watching an animation related to senior care. Such activities can induce learners’ engagement and a deeper level of curiosity and interest, especially when presented at the beginning of a lesson. The second subcategory, Relevance (R), consisted of useful S-PBL scenarios related to the learners’ future jobs or the academic requirements for students of the departments of nursing and dental hygiene so that the participants could appreciate the necessity of the learning and value of the course content. In the third subcategory, Confidence (C), learners participated in S-PBL in small groups to allow them opportunities to practice the hands-on application of the knowledge and skills that they had previously learned. This was done so that they could develop confidence through successful collaborative experiences between health professions, performing patient care, and identifying medical errors. Finally, the fourth category, Satisfaction (S), involved intrinsic reinforcement in which learners’ self-evaluations and self-reflections on the learning outcomes of the S-PBL were shared in person and in groups. Extrinsic rewards, including positive feedback from the instructor provided and stationery that was offered as a souvenir of completion of the course to all participating learners, were used to enhance their levels of satisfaction.

### 2.5. Research Procedures

Prior to participating in the S-PBL, the subjects as novice health professionals had acquired the necessary basic knowledge through the IPE course “Senior Care Convergence Clinical Practicum” (15-week/semester) at the college’s department of nursing or department of dental hygiene. The S-PBL pre-briefing session (50 min) included activities such as guidance on S-PBL and senior care, introducing students to their S-PBL group through icebreakers, and a pre-discussion of patient care. During the main S-PBL session (60 min), small group activities were undertaken in each group using an elderly care manikin (Sakamoto^®^ Koharu^®^ M100-5, Osaka, Japan) in a separate simulation room that reproduced the clinical environment. With a height of 150 cm, the manikin’s head moves in all directions (front, back, right, and left), and each joint in the shoulders, elbows, wrists, hips, thighs, knees, and ankles moves like an actual elderly body. Therefore, this manikin is useful for patient care training, such as instruction in wheelchair assistance and changing patients’ positions. The manikin’s dentures are also removable, and oral care can be performed on them. Students from the departments of nursing and dental hygiene were randomly assigned to small groups of three to four, and they then participated in S-PBL activities based on the key knowledge they had acquired through their health profession courses. 

In the small group activities, the case presented by the instructor was analyzed. The students performed basic care, such as checking vital signs and oral care, and tried to determine the risk of medical errors that posed a threat to patients’ safety, which were prepared by the instructor in advance. In the activities aimed at finding medical errors, the students participated in S-PBL to find various hazards in the patient environment based on the 2021 Patient Safety Goals presented by the Joint Commission, as follows: mismatched patient name and registration number between the patient’s ID band, IV fluid, etc. (to correctly identify the patient); prescription record and medication preparation for a drug that the patient is allergic to; medications not locked up on the nursing cart (to use medicines safely); lowered side rail of the bed (to prevent patients from falling); spilled water on the floor of the ward; wheelchair location presenting an obstacle to moving patients (to identify patient safety risks), and so on. The running time of the S-PBL per group was within 15 min, and a handover between groups was performed using the SBAR (situation, background, assessment, recommendation) format to enhance the communication between the participants and allow the subsequent group to participate in the S-PBL in sequence. The instructor ran the simulation and evaluated the students’ performance using the checklist on patient care and identifying medical errors shown in Table 1. The debriefing session (50 min) proceeded with the instructor’s feedback and the students’ self-evaluations and reflections to conclude the entire program.

### 2.6. Data Collection

This study was conducted from 3 March to 12 May 2021, after prior guidance was given to students of the departments of nursing and dental hygiene who expressed that they wished to voluntary participate in the course. To prevent the diffusion or imitation of treatments, the locations before and after the S-PBL participation were differentiated, except for the location of the pre-briefing and debriefing, so that the participants would not meet each other before or after the intervention. The pre–post surveys took about 10 min and employed a structured questionnaire. The consent form and questionnaire were collected separately so that there was no exposure of the subject’s personal information or disadvantages given by the evaluator. The researcher offered 1000 won worth of stationery to all subjects who participated in the S-PBL activity.

### 2.7. Statistical Analysis

Data analysis was performed using IBM SPSS Statistics ver. 22.0 (IBM Co., Armonk, NY, USA). The participants’ general characteristics were analyzed using descriptive statistics; the reliability of the assessment scales was assessed using Cronbach’s alpha coefficient, correlations between variables were explored by Pearson correlation coefficient, and the intervention effect was examined using a *t*-test. The correlation between factors was determined by the value of the correlation coefficient, and a *p* value < 0.05 was considered a statistically significant level.

### 2.8. Ethical Considerations

This study was approved by the Institutional Review Board of Eulji University (No. EUN20-041). As the subjects were students, the researcher provided sufficient oral explanations about the research and S-PBL in person. The participants freely expressed their wish to voluntary participate or not, and they were assured that there was no disadvantage to choosing not to participate, thereby relieving any unnecessary tension related to taking part in the research. Whenever a participant experienced a new perception or stress related to the concept of assessment in the study through completing the questionnaire, the researcher provided support with appropriate feedback and communication.

## 3. Results

### 3.1. General Characteristics

The participants’ general characteristics are shown in Table 2. The subjects mean (SD) age was 23.88 (5.20) years; 23 were female (95.8%) and one was male (4.2%). Seventeen students (70.8%) were studying nursing, and seven (29.2%) were majoring in dental hygiene; 19 (79.2%) students were in the third year of their studies, and five (20.8%) were in the fourth year. Seventeen students (70.8%) answered that they had no religion, six identified as Christian (25.0%), and one was Catholic (4.2%). In terms of health status, 22 students (91.7%) answered that they were very healthy, and two students (8.3%) answered that they were healthy, indicating that the subjectively perceived health status was good overall. As for the motivation for their choice of major, eight students (33.3%) answered that they chose their major to secure employment, and six students (25.0%) mentioned their aptitude or a recommendation from their parents or others. Fourteen students (58.3%) answered that their academic achievement was at a moderate level, and five students (20.8%) and three students (12.5%) answered that their achievement level was high or very high, respectively, indicating that the academic achievement of the students participating in the research was generally in the upper-intermediate level. Twenty-two students (91.7%) were living at home with their family. In terms of satisfaction with their majors, nine students (37.5%) answered that they were satisfied or moderately satisfied, and six students (25.0%) answered that they were very satisfied, indicating that the level of major satisfaction was generally high among the participants.

### 3.2. Correlation Between Variables

The results of examining the correlation between the variables are presented in Table 3. Among the subscales of empathy, fantasy showed mutual positive (+) correlations with empathic concern (r = 0.50, *p* = 0.01) and personal distress (r = 0.48, *p* = 0.02), and empathic concern showed a mutual positive (+) correlation with personal distress (r = 0.49, *p* = 0.02).

### 3.3. Effects of the S-PBL

The results of testing and verifying the effects of the S-PBL are outlined in Table 4. First, among subscales of empathy of the subjects, perspective taking, empathic concern, and personal distress showed an increase after participation in the S-PBL compared to the values before participation; however, the difference was not significant, and Hypothesis 1 was, therefore, rejected (*p* > 0.05). Second, the subjects’ attitudes toward the elderly showed a significant increase (t = 3.11, *p* = 0.01) after participating in the S-PBL compared to the values before participation, which supported Hypothesis 2. Third, the subjects’ team efficacy showed a significant increase (t = 2.84, *p* = 0.01) after the S-PBL participation compared to the values before; thus, Hypothesis 3 was supported.

## 4. Discussion

This study investigated the effects of a course featuring S-PBL on the empathy, attitudes toward caring for the elderly, and team efficacy of undergraduate health profession students in terms of their learning transfer. The results demonstrated that the S-PBL had a significant effect on the undergraduate students’ attitudes toward caring for the elderly and team efficacy. 

The S-PBL developed in this study was designed based on a female elderly patient case scenario, in which the health profession undergraduates collaboratively performed patient care and considered and identified medical errors to reduce hazards in the patient environment. According to the curricular theme and learning objectives of the S-PBL, the students’ attitudes toward caring for the elderly showed a significant improvement, in terms of changes in their mindsets or attitudes toward the patient receiving health care services after the S-PBL participation compared with before. The subjects’ attitudes toward caring for the elderly improved by 2.50 (3.93) points from the pre-mean (SD) of 64.54 (9.58) to the post-mean (SD) of 67.04 (11.05). This is a similar level of improvement as was reported in Joung and Hyun’s study [21] with elderly care facility employees, which found an improvement of 2.61 (−1.4) from the pre-mean (SD) of 69.11 (6.12) to the post-mean (SD) of 71.72 (4.72). In Joung and Hyun’s study [22], the control group showed a decrease of 3.45 (−0.13) points when comparing the pre- and post-mean (SD) scores of their attitudes toward caring for the elderly. It is thought that the senior simulation program of this previous study and the experiential learning of the S-PBL as interventions of the present study can contribute to broadening students’ perspectives toward caring for elderly patients and their understanding of patient care. In addition, it is useful to implement these educational interventions in the basic curriculum, starting from the undergraduate course, as well as in on-the-job training and continuing education for health professionals.

South Korea is expected to enter a super-aged society, with 15.7% of the total population being age 65 and over in 2020, and 20% predicted to be 65 and over in 2025. The life expectancy of the population aged 65 and over is a further 20.8 years (18.7 years for men, 22.8 years for women), which is 0.5 years higher for men and 1.5 years higher for women, than the OECD average [28]. Elderly patients are often bedfast due to cognitive declines, such as dementia and various chronic diseases, leading to a high level of reliance on health care services including nursing care. They are also subject to high risks of bedsores or falls, requiring highly intensive, hands-on health care services [29]. The curricular theme of this study’s S-PBL, which reflects the accelerating pace of the aging population in Korea, is highly relevant to major issues in the public health field, as well as students’ future job or academic requirements. As such, this is thought to have promoted students’ motivation, based on the Keller’s ARCS model [26]. For future health professionals, including nurses and dental hygienists, changes in the university curricula are necessary to accommodate the changes of the times and the social demands of the future, and it is essential to include related themes within the curricula of these majors. In particular, attitudes toward caring for the elderly among health professionals is one of the key factors that can improve the health outcomes of patients receiving health care services. However, due to the COVID-19 pandemic, undergraduate health profession students currently have difficulties in securing experiential learning in clinical settings [7]. Therefore, based on the situations and cases that are frequently encountered in clinical settings, practical and integrated clinical education such as S-PBL is highly useful for improving and enhancing students’ learning attitudes and critical thinking [19].

The present study observed a significant improvement in team efficacy among the undergraduate students who participated in the S-PBL compared to their efficacy before participation. Based on Keller’s ARCS model [26], S-PBL induces students’ learning motivation and is effective in promoting students’ active participation in learning in terms of learning transfer through S-PBL’s team-based nature, in contrast to individual-oriented learning. Since S-PBL is a teaching method that combines PBL based on a simulation that enables practical experiential learning of knowledge, skills, attitudes, and cases that students may encounter in clinical practice, it is believed that the S-PBL contributed to the promotion of learning transfer among the students. Although team-based learning is gradually increasing in the curriculum of universities cultivating future talents, there are still many cases where evaluation is based primarily on individual academic achievement, and, thus, it is necessary to assess the performance of the team, not the individual, during team activities [25]. Furthermore, undergraduate health profession students learn with, from, and about each other to improve teamwork, collaboration, and the quality of care through interactive learning based on IPE [6]. In this regard, based on the present study’s results, we propose that the effectiveness of S-PBL in various health profession university courses should be verified by considering the motivational strategies and tactics related to students’ attention, relevance, confidence, and satisfaction. Future research should develop various S-PBL methodologies and verify their effects in comparison with conventional lecture-oriented classes for students in various health profession disciplines.

Although the study did not confirm a significant difference in the levels of students’ multidimensional constructs of empathy, which are considered key components in theoretical and practical perspectives [29], perspective taking, empathic concern, and personal distress, all increased. Specifically, perspective taking refers to the tendency to take another person’s psychological standpoint or attitude; empathic concern refers to the tendency to feel sympathy, share pain, and take interest in others who are in distress; personal distress refers to the tendency to feel discomfort and share others’ misfortune or pain [18]. Teaching about and practicing empathetic behaviors is a helpful construct in health profession education [15]. As the ability to form empathy with other health professionals providing patient care is an indispensable factor in improving clinical outcomes, the development and application of various learning strategies in undergraduate courses that can enhance empathy and verifying the effectiveness of these strategies has significant implications [15]. However, this study experienced some practical difficulties that should be considered. First, the students experienced S-PBL for the first time and the S-PBL was only applied for a short period of time on the COVID-19 pandemic. Second, only undergraduate students studying nursing and dental hygiene participated, and undergraduates of other health profession disciplines did not. Finally, there was a lack of opportunities for hands-on training in patient care skills before the integrated implementation of the S-PBL. In consideration of these aspects, we propose that subsequent studies should consider factors that impede the engagement with, and concentration of, S-PBL and those that promote empathy in order to examine the learning transfer effects of these factors.

### Limitations

This study was conducted with students of the nursing and dental hygiene departments of a college specializing in health care and has some limitations in terms of the small sample size and lack of comparison group due to its one-group experimental design. In addition, the effect of the S-PBL in terms of learning transfer was evaluated via self-assessment rather than objective measures. Therefore, careful interpretation is required when generalizing these results. Despite these limitations, the study’s findings are significant in that they highlight the need for the development and application of IPE-based S-PBL as a novel teaching method for various health profession undergraduates, including nursing students, and present a foundational basis for follow-up studies to further verify the effectiveness of the developed method.

## 5. Conclusions

It is an exploratory study aimed at developing and applying innovative teaching methods for promoting learning transfer via S-PBL and verifying the effectiveness of the developed educational strategies. Given the special circumstances resulting from the recent COVID-19 pandemic, it is difficult to expect experiential learning effects because undergraduate health profession students, including those studying nursing and dental hygiene, have limited experience of hands-on health care service in clinical settings. The COVID-19 pandemic has necessitated alternative strategies to maintain high-quality education and to have continuous interaction and collaboration. To address this difficulty and promote learning transfer for understanding, communication, and cooperation among health professions within the university’s curriculum, this study developed and applied integrated simulations based on clinical practice situations and verified the effects of the IPE-based S-PBL. In particular, the study investigated the effects of the S-PBL in terms of learning transfer on empathy, attitudes toward caring for the elderly, and team efficacy. The study’s findings present an instrumental basis for integrated learning between health professions and an alternative teaching/learning methodology that complements the limitations in the clinical practicum of health profession undergraduates.

## Figures and Tables

**Figure 1 ijerph-18-09658-f001:**
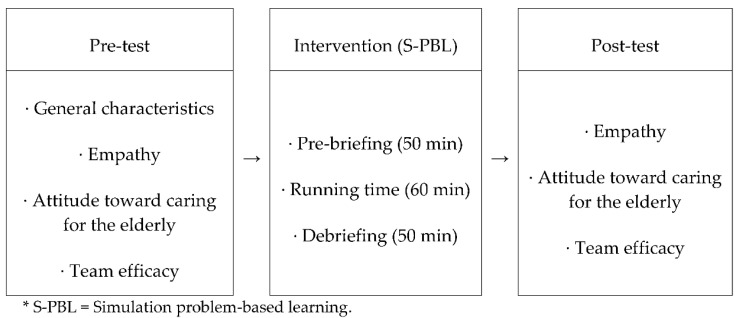
Research design.

**Figure 2 ijerph-18-09658-f002:**
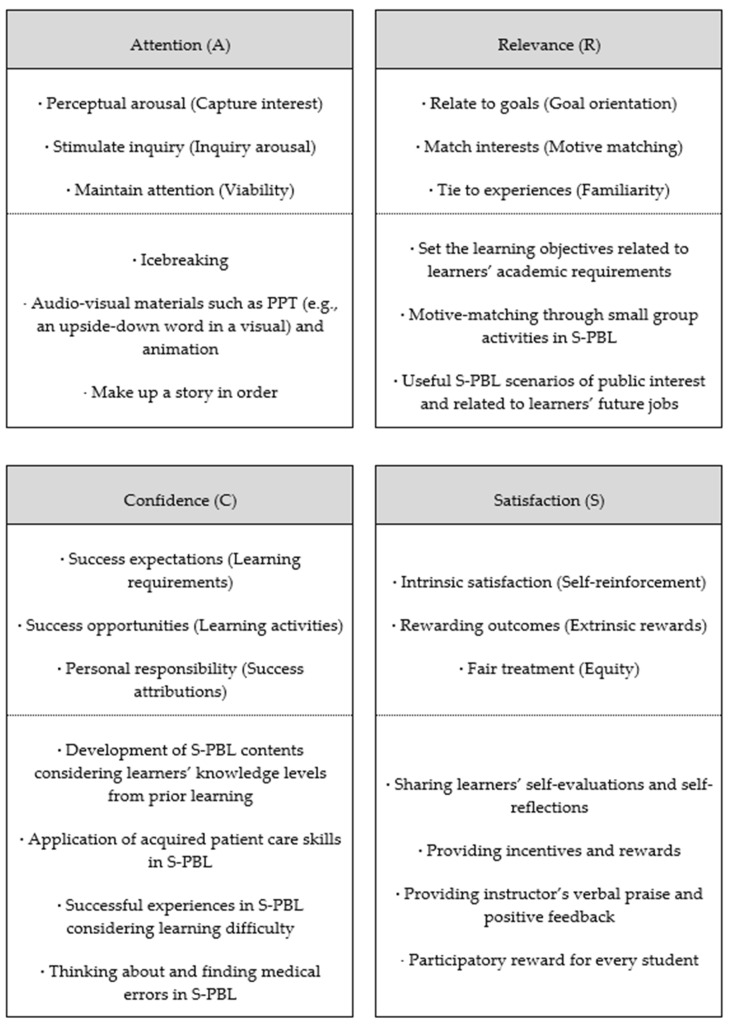
Motivational tactics in the S-PBL based on subcategories of the ARCS model.

**Table 1 ijerph-18-09658-t001:** S-PBL Checklist.

Items
Perform handwashing
Introduce yourself to the patient and double-check the patient’s identifiers.
Perform appropriate patient care.
1. Check vital signs.
2. Perform salivary gland massage for xerostomia.
3. Perform oral care, etc.
Improve the effectiveness of communication among team members (SBAR).
Record overall patient condition and care performed.
Check for medical errors.
1. Different registration numbers in different places such as patient ID band, bed name tag, and nursing record sheet.
2. Bed side rails are down, bed wheels not fixed (fall risk).
3. Patency is not maintained with the indwelling catheter clamped.
4. Urinary bag is tied to the side rail higher than the patient’s bladder.
5. The pulse oximetry sensor is not attached to the patient’s hand.
6. Water spill on the ward floor, obstructive wheelchair location in line of patient movement.
7. The patient’s allergy to antibiotics (+) is indicated in the nursing information record, but there is a doctor’s prescription and setup for the administration of the medication.
8. IV insertion (+) in the paralyzed right arm.
9. IV site clot (blood reflux and IV line full of air bubbles).
10. Uncapped 3-way stopcock (IV).
11. A different patient’s name tag is attached, and IV infusion is performed with IV fluid different from the doctor’s prescription.
12. The medication bottle is placed on the nursing cart with the lid open (not locked up).
13. The medicine for refrigerated storage is left at room temperature on the nursing cart.
14. Uncapped needles with “blood samples” (no blood test bar code attached).
15. Urine samples left at room temperature (no urinalysis bar code attached).
16. Foot drop, a roll of tape underneath the heel, and a wet patient’s clothes and sheets (bed sore risk).
17. A coffee cup used by others is placed on the tray table.
18. Empty hand sanitizer (difficult to perform hand hygiene).
19. Needle box is full and in need of replacement (risk of needle injury).
20. Oral wet gauze covers the nose and mouth of a patient complaining of xerostomia (patient safety risks, increase in the risk of aspiration, asphyxia, etc.).

**Table 2 ijerph-18-09658-t002:** General characteristics (N = 24).

Characteristics	Categories	Mean (SD)/Frequency (%)
Age (years)		23.88 (5.20)
Gender	Female	23 (95.8)
	Male	1 (4.2)
Major	Nursing	17 (70.8)
	Dental hygiene	7 (29.2)
Grade	Third year	19 (79.2)
	Fourth year	5 (20.8)
Religion	Christian	6 (25.0)
	Buddhist	0 (0.0)
	Catholic	1 (4.2)
	None	17 (70.8)
Health status	Very healthy	22 (91.7)
	Healthy	2 (8.3)
	Moderate	0 (0.0)
	Unhealthy	0 (0.0)
	Very unhealthy	0 (0.0)
Motivation for major choice	Employment	8 (33.3)
	Aptitude	6 (25.0)
	Recommendation from others	1 (4.2)
	High school grades	6 (25.0)
	Occupational image	1 (4.2)
	Other	2 (8.3)
Academic achievement	Very high	3 (12.5)
	High	5 (20.8)
	Moderate	14 (58.3)
	Low	2 (8.3)
	Very low	0 (0.0)
Residential type	Living with family	22 (91.7)
	Living alone	2 (8.3)
Satisfaction with major	Very satisfied	6 (25.0)
	Satisfied	9 (37.5)
	Moderate	9 (37.5)
	Unsatisfied	0 (0.0)
	Very unsatisfied	0 (0.0)
Empathy	Perspective taking	25.67 (3.41)
	Fantasy	24.71 (5.58)
	Empathic concern	26.54 (2.99)
	Personal distress	19.92 (5.68)
Attitudes toward caring for the elderly		64.54 (9.58)
Team efficacy		33.83 (4.23)

**Table 3 ijerph-18-09658-t003:** Correlation of variables (N = 24).

Variables	r (*p*)					
	1	2	3	4	5	6
Empathy						
1. Perspective taking	1	0.40 (0.06)	0.39 (0.06)	0.29 (0.17)	0.06 (0.77)	0.08 (0.70)
2. Fantasy		1	0.50 (0.01)*	0.48 (0.02) *	0.12 (0.58)	0.04 (0.85)
3. Empathic concern			1	0.49 (0.02) *	0.19 (0.38)	0.28 (0.19)
4. Personal distress				1	0.08 (0.72)	0.05 (0.82)
5. Attitudes toward caring for the elderly					1	0.32 (0.13)
6. Team efficacy						1

* *p* < 0.05.

**Table 4 ijerph-18-09658-t004:** Effects of S-PBL (N = 24).

Variables	Mean (SD)			t (*p*)
	Pre-Test	Post-Test	Differences	
Empathy				
Perspective taking	25.67 (3.41)	26.13 (3.87)	0.46 (2.32)	0.97 (0.34)
Fantasy	24.71 (5.58)	24.08 (6.19)	−0.63 (2.12)	−1.44 (0.16)
Empathic concern	26.54 (2.99)	27.00 (3.92)	0.46 (2.34)	0.96 (0.35)
Personal distress	19.92 (5.68)	20.63 (6.40)	0.71 (3.14)	1.11 (0.28)
Attitudes toward caring for the elderly	64.54 (9.58)	67.04 (11.05)	2.50 (3.93)	3.11 (0.01) *
Team efficacy	33.83 (4.23)	36.38 (4.23)	2.54 (4.38)	2.84 (0.01) *

* *p* < 0.05.

## Data Availability

The data presented in this study are available on request from the corresponding author. The data are not publicly available due to restrictions e.g., privacy or ethical.

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
