# Peer review of "The Effects of Simulation Problem-Based Learning on the Empathy, Attitudes toward Caring for the Elderly, and Team Efficacy of Undergraduate Health Profession Students"

_ijerph, 2021, doi:10.3390/ijerph18189658_

Round 1

Reviewer 1 Report

The manuscript contributes in an area where there is a need to increase the evidence available. 

Thank you for this paper which presents an interesting topic of issues relevant to IPE.  A stronger link both in the introduction to the paper as well as in your discussion that ties in previous research on the role of interprofessinal in intervation factors that impact empathy and attitudes toward caring for the elderly would certainly be helpful.

Author Response

Thank you for your considerate opinions. Each review opinion has been fully considered, as shown below, to improve the quality of our research paper.

I added a description on this matter in the “1. Introduction” and “4. Discussion” section. In particular, this study is meaningful for having demonstrated the use of IPE-based S-PBL in improving learning transfer in various health profession disciplines during the COVID-19 pandemic. The ARCS model of motivation proposed by Keller is a useful model for nursing simulation education. The reference list, overall, has been reviewed and revised.

I appreciate your recommendation. If anything else is required with regard to this, I will be glad to consider the same. I hope you are staying healthy and safe in the midst of the COVID-19 pandemic. Thank you.

Reviewer 2 Report

Dear Authors, I found your work interesting and valuable for a publication. Here you'll find some concerns hoping you'll find it useful to improve your MS. Lines 31 to 34: you must circumscribe this kind of preparation (you wrote ‘uni-professional education’). Please, specify that you're referring to your specific context. Lines 35 to 37: are you referring to a new and further condition that may negatively impact the health professionals' education? You proposed the S-BL as one of the demonstrated effective methods to prepare these professionals. However, before your quasi-experiment, what kind of pedagogical method this population (i.e., nurses) received? You briefly explained this point at lines 159-161, but I suggest introducing that at this point to make the central core of the current research. Before introducing your aims, it would be better to report studies adopting IPS-based S-BL with health professions. This point could better support the original contribution of your research. Concerning participants, you should add more info regarding their pre-study preparation. Are they at the beginning of their training? Are they similar in this regard? And so on. To me, it is not clear why you spent so many details on participants’ characteristics, considering that you didn’t may use it in your analysis (and hypothesis, of course).

Author Response

We deeply appreciate all of your valuable opinions. We have fully considered the reviewers’ opinions to improve the quality of our research paper.

In particular, this study is meaningful for having demonstrated the use of IPE-based S-PBL in improving learning transfer in various health profession disciplines during the COVID-19 pandemic. The ARCS model of motivation proposed by Keller is a useful model for nursing simulation education. After careful consideration of the feedback, further explanations have been added to the main text to enhance readers’ understanding. The reference list, overall, has been reviewed and revised.

If anything else is required with regard to this, I will be glad to consider the same. I hope you are staying healthy and safe in the midst of the COVID-19 pandemic. Thank you for your considerate opinions.